# Commonalities and differences in injured patient experiences of accessing and receiving quality injury care: a qualitative study in three sub-Saharan African countries

Ciaran Kennedy,[1] Agnieszka Ignatowicz  ,[1] Maria Lisa Odland,[1,2,3,4,5] Abdul-Malik Abdul-Latif,[1,6] Antonio Belli,[7,8] Anthony Howard,[9,10] John Whitaker,[1,11] Kathryn M Chu,[12,13] Karen Ferreira,[12] Eyitayo O Owolabi,[12] Samukelisiwe Nyamathe,[12] Stephen Tabiri,[14,15,16] Bernard Ofori,[14] Sheba Mary Pognaa Kunfah,[14] Mustapha Yakubu,[15,17] Abebe Bekele,[18,19] Barnabas Alyande,[20,21] Pascal Nzasabimana,[22] Jean-Claude Byiringiro,[22,23] Justine Davies[1,12,24]

**Correspondence to**
Dr Agnieszka Ignatowicz;
a.m.ignatowicz@bham.ac.uk

## ABSTRACT

**Objectives** To understand commonalities and differences in injured patient experiences of accessing and receiving quality injury care across three lower-income and middle-income countries.

**Design** A qualitative interview study. The interviews were audiorecorded, transcribed and thematically analysed.

**Setting** Urban and rural settings in Ghana, South Africa and Rwanda.

**Participants** 59 patients with musculoskeletal injuries.

**Results** We found five common barriers and six common facilitators to injured patient experiences of accessing and receiving high-quality injury care. The barriers encompassed issues such as service and treatment availability, transportation challenges, apathetic care, individual financial scarcity and inadequate health insurance coverage, alongside low health literacy and information provision. Facilitators included effective information giving and informed consent practices, access to health insurance, improved health literacy, empathetic and responsive care, comprehensive multidisciplinary management and discharge planning, as well as both informal and formal transportation options including ambulance services. These barriers and facilitators were prevalent and shared across at least two countries but demonstrated intercountry and intracountry (between urbanity and rurality) variation in thematic frequency.

**Conclusion** There are universal factors influencing patient experiences of accessing and receiving care, independent of the context or healthcare system. It is important to recognise and understand these barriers and facilitators to inform policy decisions and develop transferable interventions aimed at enhancing the quality of injury care in sub-Saharan African nations.

## STRENGTHS AND LIMITATIONS OF THIS STUDY

⇒ The data collection for this study was conducted during periods of national COVID-19 lockdowns in Ghana, South Africa and Rwanda, which may have affected the quality and depth of the data.
⇒ Only patients with musculoskeletal injuries were included in the study sample.
⇒ We employed multiple analysts and validated our findings by triangulating them with existing literature to increase the credibility of our results.

## INTRODUCTION

Approximately 90% of injury-related mortality occurs in low-income and middle-income countries (LMICs).[1] Across the African continent, the past two decades have witnessed a near 50% rise in healthy life-years lost from road traffic incidents.[1] This increased prevalence of injury has significant implications for health system capacity and finances.[2] While preventative measures are urgently needed to reduce injury prevalence, access to quality healthcare for injured individuals is also essential to save lives and prevent disability.[3 4] Unfortunately, the past 25 years have seen a critical underfunding of injury care, compared with other global health concerns.[5] The establishment of a Sustainable Development Goal target (3.6) to halve global deaths and injuries from road traffic accidents by 2030 may help to redress this balance.[6]

Ensuring equitable access to high-quality care following an injury is essential step

in improving health outcomes in LMICs.[7 8] Adopting a comprehensive approach that tends to the needs of injured individuals from the moment of injury until optimal rehabilitation can improve both mortality and morbidity rates.[4] However, knowledge of barriers to accessing care after injury is limited[9] and has mostly been studied at the healthcare facility level; studies seldom look across the whole patient pathway, from point of injury to discharge from care.[7] Those studies that have explored the whole pathway have identified multiple barriers, with the majority (around 60%) focusing on barriers to receiving quality care.[8] Identifying how to increase equitable access to injury care will require a whole health system approach to injury research, not one solely focused on the capacity for patients to receive acute care.[7–9]

Exploring patient experiences of injury and injury care along the whole care pathway can contribute towards a systems-level understanding of injury care. Using qualitative methodology enables movement beyond simply delineating barriers in access to services, to assessing the acceptability of services and subjective facilitators of quality injury care.[8] It also has the potential to elicit contextually dependent solutions to access barriers. Exploring these perspectives across multiple LMICs and foregrounding their commonalities may afford opportunities to improve health across multiple nations through transferable interventions. Therefore, our aim in this paper was to understand commonalities and differences in injured patient experiences of accessing and receiving quality injury care across three lower-income and middle-income countries.

## METHODS

This analysis used qualitative data collected from the Equi-Trauma study conducted in Ghana, South Africa and Rwanda. It is reported according to the Consolidated criteria for Reporting Qualitative research checklist.[10]

### Participants and sampling strategy

Methods for the main study have been published elsewhere.[8] For the main study, in each country, we aimed to recruit 10 participants in both the urban and rural areas to ensure a relatively equal number of patients. Convenience sampling was employed to select persons in communities who had been injured and accessed—or attempted to access—care for an injury which occurred in the previous 6 months. Where possible, purposive sampling was done to enable representation of individuals across ages, sexes and injury types. Hospital lists and contact with community leaders were used to identify participants who had or had not presented to clinics or hospitals for their injuries. When injured patients were not able to participate (eg, after death or disability), friends or relatives present during the injury were invited to participate. For this paper, we only included participants who had a musculoskeletal injury as this was the main mechanisms of injury across the four study countries.

### Data collection process

Face-to-face in-depth interviews lasting up to an hour were conducted between January and October 2021 in urban and rural settings in all three countries. Interviews were conducted in the participants' preferred language and at locations suitable for participants, including hospitals, clinics or patients' homes. Participants were interviewed by a male and female surgeon in Ghana (SMPK and MY), a male non-clinical researcher in Rwanda (PN), and two female junior doctors in South Africa (KF and SN), all native to the country in which data were collected. If interviews were not conducted in English, translation was conducted by in-country researchers (KF, MY, PN, SN and SMPK) and checked by in-country investigators (KMC, J-CB, ST).

### Data collection tools

All interviewers were trained by a senior member of the research team (AI) to ensure standardisation of data collection methods. The English version of the topic guide, before adaptation to local contexts, is attached (online supplemental appendix 1). This was developed to enable a rich understanding of patients' experiences of injury care along the whole care pathway from initial injury to rehabilitation. The topic guide was based on the authors' experiences and knowledge of the literature, permitting interviewer discretion to explore themes that emerged in specific interviews. It was piloted, and interviewers received feedback from a qualitative methods expert (AI) on their interviewing technique and field notes after two preliminary interviews. In all three countries, data saturation was reached towards the last interviews, reflecting the complexity of contexts and richness of the data.

### Analytical approach

Interviews were audio recorded and then transcribed verbatim. Computer-assisted thematic content analysis was conducted using QSR NVivo.[11] Codes were generated inductively, focusing on perceived barriers and facilitators to accessing quality injury care. A sample of interview transcripts across three study countries was read to identify the initial set of codes by two coauthors (AI and CK) through serial discussions. This generated an initial coding framework that was discussed in an analysis meeting between members of the research team (AI, CK and JD) and then used to code all remaining interview transcripts. Codes were gradually built into broader categories, with final themes being selected through comparison across transcripts and through discussion among coauthors, resolving discrepancies. Barriers and facilitators were coded as such dependent on how the participant reported them. Hence, themes could have been reported both as a barrier and a facilitator, depending on whether the participants reported them negatively or positively. Some themes with perceived substantial thematic overlap were paired as a result of the further discussions between

the two main researchers (AI and CK). Paired themes were reported together.

Barriers and facilitators that were shared across at least two countries were identified, as well as discordant themes that were prevalent in only one country. The patient-reported barriers and facilitators to good experience of trauma care for each country are available in appendices (online supplemental appendices 2 and 3). We then corroborated our results with the findings of our prior mixed-methods study in Ghana, Rwanda and South Africa.[8]

### Description of sample

A total of 60 in-depth interviews were conducted across Ghana, Rwanda and South Africa, evenly segmented by urban versus rural areas, as part of the main study. Ghanaian interviews were held in the Northern regional capital of Tamale and rural Yendi district (3 female and 17 male injured individuals). Rwandan interviews were held in the capital city of Kigali and rural Burera district of the Northern Province (5 female and 15 male injured individuals). South African interviews were held in the urban township Khayelitsha and across the rural Western Cape (4 female and 16 male injured individuals). Most of the persons interviewed in the main study had experienced musculoskeletal injuries and were eligible for inclusion in this study. For this paper, we only included participants who had a musculoskeletal injury as this was the main mechanisms of injury across the four study countries.

### Patient and public involvement

Patients and/or the public were not involved in the design, reporting or dissemination plans of this research.

### Authors' reflexivity statement

The Equi-Trauma Collaborative was a research team comprising academics, researchers and clinicians based in the UK, Rwanda, Ghana and South Africa. The team members had diverse cultural backgrounds. CK, AI, JD, MLO, ABell, AH and JW represented high-income country researchers. The knowledge of lead investigators (KMC, JC and ST) and involvement of local researchers in each study country have been crucial to the development and success of this project. All research staff who engaged in data collection are acknowledged as authors; specific roles of authors are outlined in this methods section and acknowledgements. The NIHR grant that funded this project was used to support the local research teams to undertake data collection, purchase qualitative analytical software and disseminate the findings. Multiple virtual meetings were held with all research partners to train staff to undertake data collection, analyse the data and write up the results. There are three other manuscripts published as a result of this research, two of which are first authored by the early career researchers from Rwanda and South Africa.[12 13] The manuscript from Ghana is currently in preparation.

The findings from this research formed the basis of an application for a larger grant that has been successful in securing the funding (Equi-injury: NIHR Global Health Group on Equitable Access to Quality Health Care for Injured People in Four Low or Middle Income Countries). The project is co-led by the PIs based in the UK and South Africa. With small changes to the team in the UK and addition of another partner country (Pakistan), we have formed the Equi-injury Group. There is a large capacity building component attached to this project, which focuses on building sustainable partnerships and South-South learning.

## RESULTS

All individuals approached agreed to participate in the study with an attrition rate of 0%. Several themes were paired together due to perceived substantial thematic overlap between them, a decision reached through discussions between researchers (AI and CK) during the analysis and interpretation of findings stage. These were 'individual financial scarcity' and 'inadequate health insurance coverage'; 'empathetic care' and 'responsive care', and 'information giving' and 'informed consent' . Some themes were aggregated (summarised and reported together to provide an overall view because of the need to keep the Results section condensed) and included 'health literacy' and 'information giving', 'informal transportation' and 'ambulance transportation', 'multidisciplinary management' and 'discharge planning'.

A total of 34 unique barriers and 25 unique facilitators to accessing perceived quality injury care were identified across the three countries (tables 1 and 2). In Ghana, there were 11 barriers and 14 facilitators; in Rwanda, 22 barriers and 21 facilitators; and in South Africa 23 barriers and 19 facilitators. Barriers and facilitators by rural or urban setting per country are presented in appendices (online supplemental appendices 4–9). Direct quotations are shown for the most prevalent barriers common to each country and the most prevalent facilitators common to each country.

### Common barriers

Five barriers were shared and prevalent across at least two countries (table 1). Less prevalent barriers are reported with their thematic frequencies in appendices (online supplemental appendices 4–6).

#### Service and treatment availability

This broad theme was coded 16 times in Ghanaian interviews, 16 times in Rwandan interviews and 12 times in South African interviews.

Some respondents noted an absence of adequate care in ambulances. On arrival at hospital, a lack of acute care beds and the lack of capacity to see patients in a timely manner was noted. Slow initial reviews were reportedly contributed to by inadequate staffing levels (particularly during religious and cultural festivals) and

**Table 1** Perceived barriers to a good patient experience of trauma care across Ghana, Rwanda and South Africa

| Barriers | All countries | | |
|---|---|---|---|
| | Total number | Rural | Urban |
| Service and treatment availability | 44 | 19 | 25 |
| Financial scarcity and inadequate healthcare coverage | 42 | 22 | 20 |
| Transportation barriers | 38 | 18 | 20 |
| Apathetic care | 29 | 11 | 18 |
| Inadequate information giving | 27 | 14 | 13 |
| COVID-19 | 19 | 7 | 12 |
| Cultural value of alternative services | 13 | 8 | 5 |
| Limited health literacy | 13 | 5 | 8 |
| Inadequate discharge planning | 9 | 4 | 5 |
| Bystander effect | 6 | 3 | 3 |
| Low population density | 5 | 4 | 1 |
| Inadequate administration of follow-ups | 4 | 2 | 2 |
| Inadequate analgesia | 4 | 2 | 2 |
| Quality of ambulances | 3 | 3 | 0 |
| Narratives of self-blame and individualising responsibility | 3 | 3 | 0 |
| Inability to self-care/loss of autonomy | 3 | 2 | 1 |
| Delayed presented due to perceived low severity | 3 | 2 | 1 |
| Suboptimal clinical management | 3 | 1 | 2 |
| Absence of care quality assessment | 3 | 0 | 3 |
| Poor communication with family | 2 | 1 | 1 |
| Absence of occupational health assessment | 2 | 2 | 0 |
| Unresponsiveness to formal complaints | 2 | 0 | 2 |
| Inadequate handovers | 2 | 0 | 2 |
| Loss to follow-up | 1 | 0 | 1 |
| Limited specialist training | 1 | 1 | 0 |
| Diagnostic uncertainty | 1 | 1 | 0 |
| Language barriers | 1 | 1 | 0 |
| Clinician–patient power inequality | 1 | 1 | 0 |
| Hospital catchment area system | 1 | 0 | 1 |
| Care responsibilities | 1 | 0 | 1 |
| Unmet psychological needs | 1 | 0 | 1 |
| Fear of surgery | 1 | 0 | 1 |
| Obstruction | 1 | 0 | 1 |
| Ethnic inequities in health system provision | 1 | 0 | 1 |

The number refers to the number of times the barrier was reported. The most prevalent barriers are in red.

escalating patients to a more specialised hospital before providing initial management. The unavailability of essential services (such as blood transfusion and medical imaging) at certain hospitals in Ghana sometimes necessitated referrals. Other services in Ghana were reportedly not offered, including physiotherapy and occupational therapy. Respondents in Rwanda noted that some hospitals had low medicinal stock and placed the onus on patients to purchase their in-patient medications from pharmacies—sometimes situated at a distance from the hospital. Respondents in South Africa identified the lack of provision of weekend theatre lists and the tendency to postpone operations after a full day of fasting.

### Transportation barriers
This broad theme was coded 15 times in interviews from Ghana, 15 times in interviews from Rwanda and 8 times in interviews from South Africa.

Many respondents relied on the whims of passing vehicles to reach definitive care. Some respondents in rural

**Table 2** Perceived facilitators to a good patient experience of trauma care across Ghana, Rwanda and South Africa

| Facilitators | All countries | | |
| --- | --- | --- | --- |
| | Total number | Rural | Urban |
| Informal transportation | 42 | 22 | 20 |
| Information giving and informed consent | 24 | 8 | 16 |
| Ambulance | 24 | 14 | 10 |
| Health insurance | 22 | 13 | 9 |
| Empathetic responsive care | 18 | 11 | 7 |
| Discharge planning | 15 | 6 | 9 |
| Multidisciplinary management | 14 | 9 | 5 |
| Bystander initial management | 13 | 5 | 8 |
| Health literacy | 12 | 7 | 5 |
| Financial leniency/charity | 12 | 10 | 2 |
| Community or familial financial pooling | 11 | 6 | 5 |
| Private healthcare | 5 | 2 | 3 |
| Personal healthcare networks | 5 | 5 | 0 |
| Shared ownership of management plan | 5 | 5 | 0 |
| Nearness of care | 4 | 0 | 4 |
| Emergency department preparedness | 4 | 2 | 2 |
| Requesting timely care | 4 | 3 | 1 |
| Cultural competence | 3 | 0 | 3 |
| NGO support | 3 | 2 | 1 |
| Provision of mobility aids | 3 | 3 | 0 |
| Communications network | 2 | 1 | 1 |
| Familial rehabilitative support | 2 | 1 | 1 |
| Patient autonomy | 2 | 1 | 1 |
| Preferential treatment | 1 | 1 | 0 |
| Police-facilitated transport | 1 | 1 | 0 |

The number refers to the number of times the barrier was reported. The most prevalent barriers are in red.

Ghana noted this opportunistic transport seldom arrived due to the low population density while some respondents in Rwanda stated that it was not feasible to use opportunistic motorcycles due to the degree of injury. Transportation was sometimes further delayed for Rwandan respondents due to the requirements of thorough police investigations.

There was consensus among most urban and rural Ghanaian respondents that there were no ambulances available for use while Rwandan and South African respondents noted ambulances were often inexplicably delayed. Respondents in urban South Africa reported their area was not serviced by ambulances due to impenetrably dense shacks and/or ambulance staffs' fears of robbery. Rural South African respondents noted that ambulances operated like a taxi services, collecting multiple patients from the scene of each injury before reaching definitive care.

Many patients were referred to multiple facilities from low to highly specialised centres for treatment until an appropriate hospital for their injury was reached—each escalation necessitating further transport. Rwandan respondents noted the scarcity of ambulances for inter-hospital transfers. Ghanaian respondents noted that the crowdedness of public transport (notably buses) led to some individuals not seeking follow-up care. Some Rwandan interviewees reported reaching follow-up by foot due to a lack of public transport, exacerbated by COVID-19-associated road and public transport closures.

### Apathetic care

This theme was coded 8 times in Ghanaian interviews, 8 times in Rwandan interviews and 13 times in South African interviews. In Ghanaian and South African interviews, reported apathetic care skewed towards the urban sample, whereas in Rwandan interviews apathetic care skewed marginally towards rurality.

Some respondents from Ghana noted they remained unattended to by healthcare professionals for hours after arriving at the final health facility and after the initial presentation. Moreover, those needing higher level care often received no care or human interaction from

healthcare professionals prior to interfacility transfer. Respondents from Ghana also noted that doctors often delegated all patient interaction to the nursing staff. Both Ghanaian and Rwandan respondents noted that nursing staff were generally inattentive to patients in pain while respondents in South Africa stated nursing staff were unresponsive to patient needs throughout the night shift. Respondents in Rwanda and South Africa stated that healthcare professionals could sometimes become quarrelsome or aggressive when patients made enquiries regarding their care.

### Paired theme: individual financial scarcity and inadequate health insurance coverage

These themes were present 19 times in Ghanaian interviews, 17 times in Rwandan interviews and 6 times in South African interviews.

Participants struggled to finance transportation to hospitals (including ambulance services), medical care, pharmaceutical costs and follow-up expenses. Respondents in Rwanda noted the selling of wealth (such as livestock and arable land) to fund medical treatment while respondents from Ghana noted foregoing medical care due to financial barriers. In South Africa, unsuccessful disability and/or unemployment benefit applications resulted in financial barriers to follow-up care for some. Inadequate healthcare coverage affected numerous Ghanaian respondents who reported to be either uninsured or holding lapsed health insurance. Some Rwandan respondents reported that the tiered nature of their health insurance system provided inadequate coverage of certain healthcare services (eg, one polytrauma patient received management for their musculoskeletal injury but not their dental trauma).

### Aggregated theme: health literacy and information giving

Health literacy (as compared against authors' knowledge of health services and understood as personal characteristics and social resources needed to access, understand and use healthcare information and services) was coded 10 times in Ghanaian interviews, 3 times in Rwandan interviews and was not coded in South African interviews. Inadequate information giving was coded 10 times in Ghanaian interviews, 5 times in Rwandan interviews and 12 times in South African interviews. These themes were aggregated after coding.

Some Ghanaian respondents reported that injured patients were taken to their family homes to deliberate on next steps, rather than directly to definitive care. Health centres were often used as the first port-of-call after injury as opposed to hospitals due to their relative proximity. They reported that hospital care was reserved for those with head injuries, loss of consciousness, substantial bleeding or soft tissue injury. Other Ghanaian interviewees reported that hospitals were inappropriate for fracture management and medical intervention may lead to limb deformity or amputation.

These individuals perceived that injury patients would be spared from these complications by using local, bone-setting services.

The Ghanaian respondents also stated that clinicians did not provide them with any information regarding their diagnosis, management plan or expected fit-to-work date. Some Rwandan respondents noted no explanations were provided for their early discharge, their follow-up plan or why they had no follow-up appointments scheduled. South African respondents reported that there was no information given on how to access interhospital transportation or their self-care requirements. One South African respondent noted no explanation was offered regarding their surgical error.

### Common facilitators

Six broad facilitators were shared and prevalent across at least two countries (table 2). Less prevalent facilitators are only reported with their thematic frequencies in appendices (online supplemental appendices 7–9).

### Information giving and informed consent

These themes were paired prior to coding and were coded 4 times in Ghanaian interviews, 12 times in Rwandan interviews and 8 times in South African interviews.

Ghanaian and South African respondents noted that clinical staff informed them about their diagnosis and imaging results. Their proposed management plans, including the regularity of bandage redressing, medications indicated and required surgical procedures were also explained. Respondents from Rwanda, particularly those in an urban setting, commented on the time given to patients to deliberate on treatment decisions, together with clinicians' support of patients' refusal of treatment after initial consent. Respect of patients' preference in the decision-making process with periodic clarification of understanding was also noted among Rwandan respondents.

### Health insurance

This theme was coded 4 times in Ghana, 17 times in Rwanda and once in South Africa.

The utilisation of health insurance in Rwanda was common in interviews, whereas its utilisation in South Africa was reported once. Across all countries, health insurance was noted to increase accessibility to healthcare services as well as the range of healthcare services available. Health insurance enabled extended in-patient stay while making out-of-pocket payments manageable. In Rwanda, certain types of insurance were considered more beneficial than others, including 'Community-Based Health Insurance' and 'RAMA' (a civil service insurance scheme). Charity also played a role in insurance provision in rural Rwanda, most notably Partners in Health, which was active in the Burera area when the rural study was conducted.

## Health literacy

This theme was coded three times in Ghanaian interviews, nine times in Rwandan interviews and was not coded in South African interviews.

Rwandan respondents (in contrast to Ghanaian respondents) primarily utilised hospital over bone-setting services after acute injury, implying that this was the norm in Rwanda. Respondents noted that in cases where an individual bystander suggested traditional methods, they tended to be overruled by group consensus. One Rwandan interviewee noted that traditional healing may lead to long-term complications, which would necessitate medical management. A Rwandan respondent suggested that ambulance care was ideal as ambulances were given priority access in traffic while another eluded to the benefit of initial patient management in an ambulance.

## Paired theme: empathetic and responsive care

These themes were paired prior to coding and were coded twice in Ghanaian interviews, 10 times in Rwandan interviews and 8 times in South African interviews. One Ghanaian respondent noted that healthcare professionals displayed a gentle manner, explored patients' feelings and showed concern for their work while another noted that doctors tried to inspire their patients. Rwandan respondents noted that healthcare professionals were responsive to their needs. This included providing analgesia in response to pain, allowing a carer to stay at a patient's bedside overnight and addressing and alleviating a patient's concern about the possibility of amputation. One Rwandan respondent commented on the provision of a counsellor who would converse with patients and assess their well-being. Respondents stated that healthcare professionals did not complain of tiredness, spoke with patients throughout procedures, with one healthcare professional accompanying a patient on their hospital transfer.

South African respondents in a rural setting noted that healthcare professionals checked patients' condition regularly, communicated in an understanding manner, provided bedpans for those who struggled to reach the toilet and regularly changed sheets in response to bleeding.

## Aggregated theme: multidisciplinary management and discharge planning

Multidisciplinary management was coded twice in Ghanaian interviews, twice in Rwandan interviews and 10 times in South African interviews. Discharge planning was coded once in Ghanaian interviews, five times in Rwandan interviews and nine times in South African interviews.

One Ghanaian respondent noted that physiotherapists provided them with a 'Zimmer' frame for physical rehabilitation while a Rwandan respondent noted that a physiotherapist attended their home and built equipment to practise mobilising. Physical therapy featured in many South African interviews, with activities including resistance band training and home exercises. This involved both inpatient and outpatient therapy, often on a regular basis. One South African respondent noted they received occupational therapy input too.

Rwandan respondents in an urban setting and South African respondents in both settings mentioned discharge planning. The Rwandan respondents noted that planned follow-up appointments were arranged for patients prior to discharge. Successive appointments enabled doctors to monitor progress (assisted by imaging) and address issues through prescribing and nurse-led wound redressing. South African respondents received clinician follow-up as well as specific dates for stitch/cast removal, wound redressing and physiotherapy.

## Aggregated theme: informal transportation and ambulance transportation

These two themes were aggregated after coding. Informal transportation was coded 14 times in Ghanaian interviews, 12 times in Rwandan interviews and 16 times in South African interviews. Ambulance transportation was coded three times in Ghanaian interviews, 14 times in Rwandan interviews and 7 times in South African interviews.

Respondents from all three countries relied heavily on passing-by motorcycles, tricycles and cars to reach definitive care. Reasons for seeking this form of transportation included its ready availability, its timeliness to arrival at (and its proximity to) the site of injury, its relative affordability and local norms. Ghanaian respondents noted that injured patients' family members often used their own vehicles, borrowed vehicles or sought known drivers in the region to transport their relative to emergency care. Transportation of injured Rwandan respondents was sometimes facilitated by the police, the army or by a doctor. In rural Rwandan settings that were inaccessible to vehicles, transportation was afforded using traditional stretchers known as 'ingobyi'. Some South African respondents reported driving themselves to acute care or being transported by their employer. Informal transportation was also employed for interhospital transfers and reaching follow-up appointments. South African respondents also noted using friends, family and neighbours to reach follow-up care while participants across the three countries utilised public transportation (including bus and taxis) to reach follow-up care.

## Discordant data

Two barriers and one facilitator were identified that were prevalent in one country only. The barrier, 'cultural value of alternative services' was coded 12 times in Ghanaian interviews—7 times in rural settings and 5 times in urban settings. The barrier 'bystander effect' was coded six times in Rwandan interviews, evenly distributed between rural and urban settings. The facilitator 'shared ownership of management plan' was coded five times in rural South African interviews.

## Rural versus urban findings

Across all three countries, the most prevalent barriers in urban settings tended to be mirrored in rural settings.

Service and treatment availability was coded 20 times in urban settings and 19 times in rural settings, individual financial scarcity and inadequate health insurance coverage were coded 20 times in urban settings and 22 times in rural settings, and inadequate information giving was coded 13 times in urban settings and 14 times in rural settings. Two prevalent barriers were shared among urban and rural settings but more frequent in urban areas: transportation barriers were coded 25 times in urban settings and 18 times in rural settings while apathetic care was coded 18 times in urban settings and 11 times in rural settings.

Across all three countries, the most prevalent facilitators in urban settings were also prevalent in rural settings. Informal transportation was coded 20 times in urban settings and 22 times in rural settings, Ambulance transportation was coded 10 times in urban settings and 14 times in rural settings, Health insurance was coded 9 times in urban settings and 13 times in rural settings, Empathetic and Responsive Care was coded 9 times in urban settings and 11 times in rural settings, and Discharge planning was coded 9 times in urban settings and 6 times in rural settings. One prevalent facilitator skewed towards urbanity while another skewed towards rurality: Information giving and informed consent was coded 16 times in urban settings and 8 times in rural settings while financial leniency or charity was coded twice in urban settings and 9 times in rural settings. This latter theme involved examples of ambulances not charging for transportation, clinicians not charging acute injury care and physiotherapists not charging outpatient care for individuals with limited financial resources in Ghana and Rwanda.

## DISCUSSION

We identified five common barriers and six common facilitators affecting patient experiences of accessing and receiving high-quality injury care in Ghana, Rwanda and South Africa. The barriers encompassed issues such as limited service availability, transportation challenges, indifferent caregiving, financial constraints and inadequate health insurance coverage, alongside low health literacy and information provision. Conversely, facilitators included effective information dissemination and informed consent practices, access to health insurance, improved health literacy, compassionate and responsive care, comprehensive multidisciplinary management and discharge planning, as well as both informal and formal transportation options including ambulance services. We discuss our findings below with reference to wider literature on seeking and accessing care in LMICs.

Individual financial scarcity and the associated theme of inadequate health insurance coverage were predominant barriers across all three countries. These barriers likely contributed to poorer patient experiences of injury care by erecting barriers to seeking, reaching, receiving and remaining in healthcare.[14] They notably led to a lack of financial risk protection, which can increase the likelihood of injured patients forgoing care or opting for traditional care.[15]

Interestingly, the response of injured patients to a lack of financial risk protection appeared to vary between countries. Among Ghanaian respondents, those without the capacity to pay tended to forgo care or incur costs. Forgone care may have led to many injured Ghanaians acquiring disability amenable to healthcare—that is, potentially preventable given timely, effective injury care.[4 16] Their disability severity in turn may have affected their capacity to work, and therefore, finance future healthcare. In this vein, Mock *et al* noted that 72% of injured Ghanaian patients reported a loss of income, primarily affected by the length of ensuant disability.[17]

Similarly, while some Rwandan respondents also reported forgoing care, others resorted to extreme measures such as selling or borrowing against their assets (eg, livestock and arable land) to cover acute injury expenses, thereby jeopardising their ability to afford future healthcare interventions and potentially exacerbating disability outcomes. The situation is compounded by the limitations of community health insurance, which, despite providing coverage for certain medical expenses, fails to address additional financial burdens such as transportation costs, caregiving expenses and lost wages, as observed by Niyigena *et al*.[18] Sapkota *et al* have previously described a vicious cycle of chronic disease and poverty in Nepal. They argued convincingly that in LMICs with significant out-of-pocket expenditure for healthcare, chronic disease and poverty become mutually reinforcing across the life course.[19]

Although South African respondents in our study did not report issues of insurance inadequacy or foregoing acute care due to financial constraints, financial barriers to follow-up care were noted, suggesting that while health insurance coverage may be more prevalent in this context, challenges in accessing comprehensive care persist, particularly in rural areas. In this respect, it becomes imperative to address the systemic disparities that hinder equitable access to comprehensive healthcare, especially in underserved rural regions, where the lingering financial barriers impede timely access to care and follow-up.

Availability of transportation emerged as a common theme across the study countries. While formal ambulance systems did not enable all injury patients to reach definitive care, informal methods offered some support. Particularly in Ghana, the absence of adequate ambulance services was felt most strongly among the respondents. However, it is worth noting that the reported access to ambulance services does not seem to translate into service uptake (at least in our small sample). Interestingly, research conducted in South Africa suggests that merely increasing the number of ambulances may not substantially improve response times, echoing sentiments from urban Cape Town respondents in our study.[20] They suggested that ambulance coverage adequacy might hinge more on the perceived safety of ambulance personnel than on their sheer availability.

In contexts like rural Rwanda, where mountainous terrain renders traditional ambulances impractical, the development of community-based transportation systems, such as using traditional stretchers, could be more viable. However, it is essential to ensure that such initiatives are not merely tokenistic gestures but involve genuine community participation in health system reforms, as observed in some instances in sub-Saharan Africa. Addressing these issues is crucial for supporting injury transportation systems effectively and ensuring equitable access to care across diverse geographies.[21] Strengthening community engagement and tailoring interventions to local needs could enhance the effectiveness of transportation solutions in sub-Saharan Africa.

Once at the healthcare facility, myriad barriers pertaining to service and treatment availability were highlighted across all three countries. Respondents noted the absence of care giving and unavailability of essential services such as transfusion and imaging services. These issues were often further compounded by medicinal and dressing shortages. The absence of essential medicines in hospitals and affiliated pharmacies can contribute to increased out-of-pocket expenditure of injury patients (for both pharmaceuticals and the travel costs to dispensaries). Research suggests that 'stockouts' affect patient safety through increasing patient exposure to substandard and falsified medicines, which are estimated to form 13.6% of drug usage in LMICs.[22] They also lead to hospital underutilisation, poorer patient experiences of care and distrust of healthcare professionals.[23] While imaging and medication may ultimately be available in some hospitals and pharmacies, respectively, they may not be affordable, hence their access will be less timely, affecting patient experiences and outcomes. In this respect, the availability of services and medicines and the affordability of episodes of healthcare lie on a continuum of access barriers to quality injury care.

In all three countries, respondents highlighted concerns regarding apathetic care while instances of compassionate care were praised. The interactions with healthcare professionals profoundly shape patients' perceptions of quality of care, with research indicating that positive patient experiences are influenced by the disposition of doctors, their tone of voice, their attitudes towards patients and their attentiveness.[24] Cultural awareness is also important in injury care settings, given the diverse patient population encountered, particularly in a country like Ghana with significant internal migration. This migration, affecting 40% of the population, results in rural-to-urban migrants facing linguistic, cultural and financial challenges. Acknowledging the diversity of ways of knowing (epistemologies) across different Global South contexts is crucial, particularly regarding the preference for traditional bone-setting services in certain sub-Saharan African populations, rooted in historical practices.[25] Hence, any effort to enhance trauma systems should not marginalise traditional knowledge but rather integrate it into the broader healthcare framework. It is worth emphasising that treatment pathways of individual injury patients are complex and often involve multiple transitions between traditional and medical services.[26] It may be more appropriate therefore to consider these services as two components of a single health system with opportunities for complementary work, rather than discrete competing treatment decisions. For instance, recent research indicates that traditional bone setters are receptive to engagement with orthopaedic services, offering opportunities to mitigate complications through joint management approaches.[27] One potential avenue for enhancing access to quality injury care could involve integrating rehabilitative physiotherapy services into bone-setting centres, fostering synergistic partnerships for improved patient outcomes.

We note that some barriers were also framed as facilitators even, sometimes, by the same participants in interviews suggesting these are different sides of the same coin. This was especially the case for transportation, where informal transportation, when available is a facilitator, whereas when unavailable is seen as a barrier. Likewise for apathetic and empathetic care. Although many of these similar themes were reported more frequently as barriers than facilitators, these findings suggests that with deeper understanding of the context, it may be possible to turn the barriers into facilitators. For example, by formalising informal transport or by holding up examples of empathetic care and the benefits of this for others to learn from.

Finally, our analysis shows that when examining patient experiences, barriers and facilitators to accessing injury care are comparable across diverse contexts. The findings suggest that there are universal factors influencing patient perceptions, independent of the context or healthcare system. This underscores the importance of recognising and understanding these common barriers and facilitators to inform policy decisions and develop transferable interventions aimed at improving access to quality injury care, ultimately leading to better outcomes and quality of care throughout sub-Saharan Africa. However, such interventions require to be codesigned by and with patients.[28] By integrating patients' perspectives and experiences, interventions can be customised to meet their specific needs and preferences, leading to improved clinical outcomes and a reduction in healthcare disparities.[29 30]

## Limitations and strengths

The data collection for this study was conducted during periods of national COVID-19 lockdown, which may have transiently affected injury epidemiology. During their lockdown, South Africa introduced an alcohol ban, which limited exposure to a significant risk factor for injury.[31] The pandemic also precluded respondent validation of transcribed interviews. Moreover, we did not stratify our analyses by gender due to the small sample size of women recruited, which would have risked deanonymising participants. These limitations have gender equity implications; however, we recognise also that most injuries are suffered

by men, which is reflected by their dominance in our sample.[1] While rural–urban divisions were explored in our study, large areas of each country were left unsampled, limiting national representativeness. The absence of ethnic coding of transcripts prohibited any investigation of ethnic inequities in injury care, which are known to exist for myriad health conditions. Further research on patients' experiences of injury care employing an intersectional lens, segmenting data by gender, ethnicity and socioeconomic status, may illuminate any undocumented inequities.

A strength of our study was that its design was informed by prior stakeholder engagement in Ghana, Rwanda and South Africa.[8 32] Community leaders were also consulted prior to the study and involved in various aspects of the research. The credibility of our study has been improved through using multiple analysts and by triangulating our findings with the published literature. Investigator checking was undertaken at multiple stages throughout the thematic coding process. Our prior mixed-methods study in Ghana, Rwanda and South Africa has been used for methodological and data source triangulation. 20/34 barriers (58.8%) in our study and 13/25 facilitators (52%) were identified in this prior study (online supplemental appendix 10) strengthening the confidence in the validity of these findings.[8]

## CONCLUSION

Our analysis of patient experiences across Ghana, Rwanda and South Africa revealed consistent themes of both barriers and facilitators in accessing and receiving high-quality injury care. Financial scarcity and inadequate health insurance coverage, transportation accessibility, service and treatment availability emerged as predominant barriers across all three countries, contributing to poorer patient experiences and potentially leading to a lack of financial risk protection. Facilitators included effective information giving and informed consent practices, access to health insurance, improved health literacy, empathetic and responsive care, comprehensive multidisciplinary management and discharge planning, as well as both informal and formal transportation options including ambulance services. Our findings suggest that there may be universal factors influencing patient perceptions of quality care, regardless of the country or healthcare system and highlight the importance of understanding common barriers and facilitators to inform policy-making and develop effective transferable strategies for improving patient experiences of injury care in sub-Saharan African countries.

**Author affiliations**
[1]Institute of Applied Health Research, University of Birmingham, Birmingham, UK
[2]Department of Obstetrics and Gynaecology, St. Olavs University Hospital, Trondheim, Norway
[3]Malawi-Liverpool-Wellcome Trust Research Institute, Blantyre, Malawi
[4]Department of Public Health and Nursing, NTNU, Norwegian University of Science and Technology, Trondheim, Norway
[5]Institute of Life Course and Medical Sciences, University of Liverpool, Liverpool, UK
[6]Volta Regional Health Directorate, Ghana Health Service, Accra, Greater Accra, Ghana
[7]Institute of Inflammation and Ageing, University of Birmingham, Birmingham, UK
[8]National Institute for Health Research Surgical Reconstruction and Microbiology Research Centre, Birmingham, UK
[9]Leeds Institute of Rheumatic and Musculoskeletal Medicine, School of Medicine, University of Leeds, Leeds, UK
[10]Nuffield Department of Orthopaedics, Rheumatology and Musculoskeletal Sciences, National Institute of Health Research (NIHR) Biomedical Centre, University of Oxford, Headington, UK
[11]King's Centre for Global Health and Health Partnerships, King's College London Faculty of Life Sciences and Medicine, London, UK
[12]Department of Global Health, Centre for Global Surgery, Stellenbosch University, Cape Town, South Africa
[13]Department of Surgery, University of Botswana, Gaborone, Botswana
[14]Ghana HUB of NIHR Global Surgery, Tamale, Ghana
[15]Department of Public Health, Tamale Teaching Hospital, Tamale, Ghana
[16]Department of Surgery, Tamale Teaching Hospital, Tamale, Ghana
[17]School of Medicine and Health Sciences, University for Development Studies, Tamale, Ghana
[18]University of Global Health Equity, Kigali, Rwanda
[19]Department of Surgery, Addis Ababa University, Addis Ababa, Ethiopia
[20]Center for Equity in Global Surgery, University of Global Health Equity, Kigali, Rwanda
[21]Program in Global Surgery and Social Change, Harvard Medical School, Boston, Massachusetts, USA
[22]University of Rwanda, Kigali, Rwanda
[23]Department of Surgery, University Teaching Hospital, Kigali, Rwanda
[24]Medical Research Council/Wits University Rural Public Health and Health Transitions Research Unit, Faculty of Health Sciences, School of Public Health, University of the Witwatersrand, Johannesburg, South Africa

**Acknowledgements** We would like to acknowledge the work of our coauthor Mr Abul Malik Abul-Latif whose tireless work enabled the completion of this study. Abdul passed away in September 2023.

**Contributors** CK conducted the analyses and wrote the initial draft. JD and AI developed the idea for this analysis and supervised CK in the analyses and writing. JD, KMC, ST and J-CB developed the protocol for the main study and oversaw it. JW, ABelli and AH contributed ideas for the protocol and running of the project. MLO, AI, A-MA-L, ABekele, BA, J-CB, ST, KMC and JD oversaw the day-to-day data collection and supervised the data collectors. KF, EOO, SN, BO, SMPK, MY and PN collected data for the study. All authors have reviewed and approved the manuscript. JD is the guarantor.

**Funding** This research was funded by the National Institute for Health and Care Research (grant number 130036).

**Competing interests** None declared.

**Patient and public involvement** Patients and/or the public were not involved in the design, or conduct, or reporting, or dissemination plans of this research.

**Patient consent for publication** Consent obtained directly from patient(s).

**Ethics approval** This study involves human participants and data collection was approved by the Ghana Health Service Ethics Review Committee (GHS-ERC005/02/20); The Stellenbosch University Health Research Ethics Committee for South Africa (N20/01/010) and the Internal Research Board of the College of Medicine and Health Sciences of the University of Rwanda for Rwanda (No 050/CMHS IRB/2020). Additional approval was obtained from the Western Cape Department of Health (WC_202006_022) in South Africa, and from the respective hospitals before visiting the facilities. The overall study was approved by the University of Birmingham Research Ethics Committee, UK (ERN_20-00880). Funding was provided by the National Institute of Health Research, NIHR, award number 130036. Participants gave informed consent to participate in the study before taking part.

**Provenance and peer review** Not commissioned; externally peer reviewed.

**Data availability statement** Data are available on reasonable request. Data are available on a reasonable request from JD (email: j.davies.6@bham.ac.uk).

peer-reviewed. Any opinions or recommendations discussed are solely those of the author(s) and are not endorsed by BMJ. BMJ disclaims all liability and responsibility arising from any reliance placed on the content. Where the content includes any translated material, BMJ does not warrant the accuracy and reliability of the translations (including but not limited to local regulations, clinical guidelines, terminology, drug names and drug dosages), and is not responsible for any error and/or omissions arising from translation and adaptation or otherwise.

**ORCID iD**
Agnieszka Ignatowicz http://orcid.org/0000-0002-5863-0828

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
