## [Reviewer comments · BMJ Open]

ARTICLE DETAILS

TITLE (PROVISIONAL)	Commonalities and differences in injured patient experiences of accessing and receiving quality injury care: a qualitative study in three sub-Saharan African countries
AUTHORS	The Equi-Trauma Collaborative, The Equi-Trauma Collaborative; Ignatowicz, Agnieszka

VERSION 1 – REVIEW

REVIEWER	Julia Hews-Girard University of Calgary, Faculty of Nursing
REVIEW RETURNED	02-Jan-2024

GENERAL COMMENTS	Thank you for this interesting paper addressing an important topic. The abstract mentions injury (broadly), however, the paper appears to focus on injury secondary to motor vehicle/road accidents. This should be more clearly defined/delineated in the abstract and the paper. Clearly state the aim of the work - it is not clear if the major aim is to compare across countries, or between rural and urban areas within countries. Please clarify in the methods why inclusion was limited to individuals with musculoskeletal injuries only and if there were other inclusion/exclusion criteria for this work in particular. Please clearly state how sample size was determined to be adequate and provide a reference. Consider rewording the sentence on pg 2 lines 46-50 for clarity. pg 4 - line 46 - please state why the results of this study were compared to previous work. Please clarify the difference between paired and aggregated themes and clarify why themes were paired. Please provide a definition for health literacy in this study. Please clarify why 2 countries for "broad" themes vs all 3 - particularly if the aim is to compare across all three countries. The way in which the results are organized could be clearer - there appears to be an overall heading, then subheadings, with the overall heading appearing as a subheading (i.e. pg 12-13). Please clarify choices regarding aggregating themes (i.e. multidisciplinary care and discharge planning). The discussion appears consistent with the results - however, without a clear aim, it's unclear how these findings fit with the literature. Additionally, some of the comparisons (advanced airway maneuvers, prehospital transfer time for pregnant individuals) are not necessarily comparable to traffic accidents. This may be helped by clearly defining inclusion/exclusion criteria, defining the type/severity of injury incurred.
---

REVIEWER	Jonas Lander Hannover Medical School, Institute for Epidemiology, Social Medicine and Health Systems Research
REVIEW RETURNED	23-Jan-2024

GENERAL COMMENTS	Commonalities and differences in patient experiences of trauma systems in three sub-Saharan African countries Title / Abstract  • As someone who's not an expert in injury/trauma research, I would find it a bit difficult to understand just by reading the title what "experiences of trauma systems" is about, particularly trauma systems may not be easily understandable • "how patients experience assessing injury care" > either experience or assess • The abstract is quite short and could be enhanced in several parts, particularly some more details in the methods and some actual contents in the findings, ie description of which barriers and facilitators you found, not just that you found them • Please also revise the conclusion, ie make it more concrete > which interventions? Who can plan and conduct them? >> the same is true for "how this study affects research and policy" Introduction  • Maybe helpful to have some examples of what injury care is about? • This section is clearly written overall. However, I am not sure if in the first paragraph, there is a logic argumentation for why there are/may be barriers to injury care ("That is, addressing barriers..."). Can you give examples or arguments for this? • For the final part of the introduction, it would help if you formulate a clear research question/objective. Methods  • Data collection > uses "Interviews were.." too often • "we developed a data-informed causal loop diagram" > please explain • I don't understand the reason you give for excluding the one person out of 60? • What was your approach/aim for data saturation, i.e. why did you conduct 60 interviews? • It would be very helpful to insert examples of your questions and or main question themes to get an idea of what you asked about – maybe a tabular summary of the interview guideline? • You mention how you analysed data, but not according to which methodological approach/theory Results  • May be helpful to provide a summary of the coding scheme (without results) either in the methods or beginning of results section for a better idea of the full scheme? • How did you assess health literacy in qualitative interviews? (this is usually measured via survey) • When was a barrier "most prevalent" and when "somewhat prevalent" etc? • On the one hand, it is good to have the barriers/facilitators visualized overview of all the themes, on the other hand, I am not sure if the beginning of the results is the right place for this, because there is basically no text to it beforehand. Usually you have this kind of overviews in the middle or towards the end of the results. Have
---

	you thought about this?  • Are you sure that “unique barriers/facilitators” is the right expression because eg health literacy is both?! • At first glance, it is unclear which “common barriers” you mean because the table does not differentiate between common and less common? If there is no connection between the table and the description of common themes, and this needs explanation because readers expect further elaboration of the tables in the text. • You begin by describing a paired theme and then a list from I-III follows. It unclear what this structure refers to and why it is III and not V, while the facilitators seem to entail I-V. Please revise this structure. It is also unclear why “pared theme...” is not provided with a number, but the following subheading is. • My overall suggestion for the results section is to try reduce this from (almost) 8 pages of text to 5-6 – it is simply too much info / too long otherwise, even if this is a qualitative paper. I guess you can reduce text particularly in the description of the individual themes. • Readers will tend to look at figure one without reading the text beforehand and I am not sure if this figure is self-explanatory, particularly re where it starts, where it ends and at which point the actual injury happens / can be imagined. The 3 outer boxes (catastrophic expenditure etc) are unclear, how do they relate to the cycle? Discussion  • The very first paragraph should entail a summary with the main barriers and facilitators and a (very) short explanation of each. Now it seems unclear where this summary section ends and if you also describe facilitators and not just barriers? • Will be very helpful to insert thematic sub-headings for the first 3 pages of the discussion for greater clarity/structure Conclusion  • Would be good to, again, say what hinders and facilitates good injury care most so far and what kind of “effective strategies” you would really suggest – maybe also by considering the local situation? This can still be concise as you don’t need to make the manuscript longer.
--	--

VERSION 1 – AUTHOR RESPONSE

Reviewers’ comments	Authors’ responses
Editor’s comments	
(1) Please report whether there was any sample size calculations - how was the total of 60 interviews arrived at?	We have explained the sampling in more detail on p. 3 (line 85). There was no formal sample size calculation as this was a qualitative study. In each country we aimed to recruit 10 participants in both the urban and rural areas. We used a purposeful sampling strategy to ensure a relatively equal number of patients from each area in each study country.
2) Please move the Patient and Public involvement section to the Methods section	This section has now been incorporated into the Methods section.
(3) Please ensure that your abstract is formatted according to our Instructions for Authors:	We have formatted the abstract according

http://bmjopen.bmj.com/pages/authors/#research	to the guidelines for Authors.
(4) Please remove the 'What is already known, What this study adds, How this study might affect research...' sections. Instead, please add a section entitled 'Strengths and limitations of this study', immediately after the abstract. This section should contain up to five short bullet points, no longer than one sentence each, that relate specifically to the methods. The novelty, aims, results or expected impact of the study should not be summarised here.	The sections have now been removed. We have added the “Strengths and limitations of this study” section after the abstract.
Please revise the title to include the study design	We have revised the title to include study design: “Commonalities and differences in injured patient experiences of accessing and receiving quality injury care: a qualitative study in three sub-Saharan African countries”
Reviewer 1	
The abstract mentions injury (broadly), however, the paper appears to focus on injury secondary to motor vehicle/road accidents. This should be more clearly defined/delineated in the abstract and the paper.	Thank you for your comment. We have rewritten the abstract and added a more detailed description of the sample.
Clearly state the aim of the work - it is not clear if the major aim is to compare across countries, or between rural and urban areas within countries.	We have added our objectives to the abstract and in the introduction section (p. 3; line 68): “Our aim in this paper was to understand commonalities and differences in injured patient experiences of accessing and receiving quality injury care across three lower- and middle-income countries.”
Please clarify in the methods why inclusion was limited to individuals with musculoskeletal injuries only and if there were other inclusion/exclusion criteria for this work in particular.	Thank you for your comment. For this paper, we only included participants that had a musculoskeletal injury as this was the main mechanisms of injury across the four study countries.
Please clearly state how sample size was determined to be adequate and provide a reference.	As per our response to the comments from the Editor, we have explained the sampling in more detail on p. 3 (line 85). There was no formal sample size calculation as this was a qualitative study. In each country we aimed to recruit 10 participants in both the urban and rural areas. We used a purposeful sampling strategy to ensure a relatively equal number of patients from each area in each study country.
Consider rewording the sentence on pg 2 lines 46-50 for clarity.	Thank you. We have rewritten this sentence: “Those studies that have explored the whole pathway have identified multiple

	barriers, with the majority (around 60%) focussing on barriers to receiving quality care”.[8]
pg 4 - line 46 - please state why the results of this study were compared to previous work.	This was done to corroborate our findings and strengthen the interpretation of data. We have elaborated on this in our ‘Strengths and limitations’ section (p. 21, line 621).
Please clarify the difference between paired and aggregated themes and clarify why themes were paired.	Thank you. We have now clarified what we mean by paired and aggregated in the Results section (p.6, line 188): “All individuals approached agreed to participate in the study with an attrition rate of 0%. Several themes were paired together due to perceived substantial thematic overlap between them, a decision reached through discussions between researchers (AI & CK) during the analysis and interpretation of findings stage. These were: ‘individual financial scarcity’ and ‘inadequate health insurance coverage’; ‘empathetic care’ and ‘responsive care’, and ‘information giving’ and ‘informed consent’. Some themes were aggregated (summarized and reported together to provide an overall view because of the need to keep the Results section condensed) and included: ‘health literacy’ and ‘information giving’, ‘informal transportation’ and ‘ambulance transportation’, ‘multidisciplinary management’ and ‘discharge planning’.
Please provide a definition for health literacy in this study.	The definition has been included (p. 11, line 318).
Please clarify why 2 countries for "broad" themes vs all 3 - particularly if the aim is to compare across all three countries.	Thank you for your comment. We see how this may be confusing. We have removed the word ‘broad’ from the sentences in the Abstract, Results and Discussion sections.
The way in which the results are organized could be clearer - there appears to be an overall heading, then subheadings, with the overall heading appearing as a subheading (i.e. pg 12-13).	Thank you, this is an important point. We have reorganised the Results section to take into consideration your comment. Hope the new layout is easier to follow.
Please clarify choices regarding aggregating themes (i.e. multidisciplinary care and discharge planning)	Some themes were aggregated (summarized and reported together to provide an overall view because of the need to keep the Results section condensed) and included: ‘health literacy’ and ‘information giving’, ‘informal transportation’ and ‘ambulance transportation’, ‘multidisciplinary management’ and ‘discharge planning’. This was a pragmatic decision because of the word limit and the

	need for clarity and transparency in reporting results.
The discussion appears consistent with the results - however, without a clear aim, it's unclear how these findings fit with the literature. Additionally, some of the comparisons (advanced airway maneuvers, prehospital transfer time for pregnant individuals) are not necessarily comparable to traffic accidents. This may be helped by clearly defining inclusion/exclusion criteria, defining the type/severity of injury incurred.	We have reworked the discussion and provided more detail about the focus of the paper in the Abstract and Introduction.
Reviewer 2	
As someone who's not an expert in injury/trauma research, I would find it a bit difficult to understand just by reading the title what "experiences of trauma systems" is about, particularly trauma systems may not be easily understandable	Thank you for this comment. We have changed the title of the paper.
"how patients experience assessing injury care" > either experience or assess	Thank you, this has now been corrected.
The abstract is quite short and could be enhanced in several parts, particularly some more details in the methods and some actual contents in the findings, ie description of which barriers and facilitators you found, not just that you found them	We have rewritten the abstract to account for the comment.
Please also revise the conclusion, ie make it more concrete > which interventions? Who can plan and conduct them? >> the same is true for "how this study affects research and policy"	This part of the Abstract has also been rewritten. It now reads: "There are universal factors influencing patient experiences of accessing and receiving care, independent of the context or healthcare system. It is important to recognize these factors and understand both barriers and facilitators to inform policy decisions and develop interventions aimed at enhancing the quality of injury care in sub-Saharan African nations."
Introduction: Maybe helpful to have some examples of what injury care is about?	Thank you for the comment. We have rewritten this section to add clarity.
Introduction: This section is clearly written overall. However, I am not sure if in the first paragraph, there is a logic argumentation for why there are/may be barriers to injury care ("That is, addressing barriers...."). Can you give examples or arguments for this?	As above, we have rewritten this section.
For the final part of the introduction, it would help if you formulate a clear research question/objective.	This has been added (p. 3; line 68): "Our aim in this paper was to understand commonalities and differences in injured

	patient experiences of accessing and receiving quality injury care across three lower- and middle-income countries."
Methods: Data collection > uses "Interviews were.." too often	Thank you. We have rewritten this section to account for your comment.
"We developed a data-informed causal loop diagram" > please explain	We have now removed the figure. Please see our response to the comment below.
Sample: What was your approach/aim for data saturation, i.e. why did you conduct 60 interviews? • It would be very helpful to insert examples of your questions and or main question themes to get an idea of what you asked about – maybe a tabular summary of the interview guideline?	Thank you. We have tried to clarify this and add more detail on our sampling on p. 3 (line 85). In each country we aimed to recruit 10 participants in both the urban and rural areas. We used a purposeful sampling strategy to ensure a relatively equal number of patients from each area in each study country. For this paper, we only included participants that had a musculoskeletal injury as this was the main mechanisms of injury across the four study countries.
May be helpful to provide a summary of the coding scheme (without results) either in the methods or beginning of results section for a better idea of the full scheme? You mention how you analysed data, but not according to which methodological approach/theory	Thank you. This is provided in appendices.
How did you assess health literacy in qualitative interviews? (this is usually measured via survey)	Thank you for this comment. We have not assessed health literacy. Health literacy was a reported facilitator to accessing injury care identified in the interviews.
When was a barrier "most prevalent" and when "somewhat prevalent" etc?	This is based on the overall number of times the barrier or facilitator was reported in interviews in each country, e.g. the most prevalent barriers are the ones that were reported most across the study countries.
On the one hand, it is good to have the barriers/facilitators visualized overview of all the themes, on the other hand, I am not sure if the beginning of the results is the right place for this, because there is basically no text to it beforehand. Usually you have this kind of overviews in the middle or towards the end of the results. Have you thought about this?	Thank you for the comment. We have chosen to report the tables with barriers and facilitators at the beginning of the Results section as they support a summary of the overall findings and set the scene for the more detailed results.
Are you sure that "unique barriers/facilitators" is the right expression because eg health literacy is both?!	Thank you for this comment. We have addressed this in the Methods section (p. 4, line 131): "Barriers and facilitators were coded as such dependent on how the participant reported them. Hence, themes could have been reported both as a barrier and a facilitator, depending on if the participants reported them negatively or positively" and in the Discussion (p. 21, line

	641): “We note that some barriers were also framed as facilitators even, sometimes, by the same participants in interviews suggesting these are different sides of the same coin. This was especially the case for transportation, where informal transportation, when available is a facilitator, whereas when unavailable is seen as a barrier. Likewise for apathetic and empathetic care. Although many of these similar themes were reported more frequently as barriers than facilitators, these findings suggests that with deeper understanding of the context, it may be possible to turn the barriers into facilitators. For example, by formalising informal transport or by holding up examples of empathetic care and the benefits of this for others to learn from.”
At first glance, it is unclear which “common barriers” you mean because the table does not differentiate between common and less common? If there is no connection between the table and the description of common themes, and this needs explanation because readers expect further elaboration of the tables in the text	Thank you for your comment. The tables with facilitators and barriers are colour coded. The number refers to the number of times the barrier was reported. The most prevalent (most common) barriers are in red. The explanation is in the legend.
You begin by describing a paired theme and then a list from I-III follows. It unclear what this structure refers to and why it is III and not V, while the facilitators seem to entail I-V. Please revise this structure. It is also unclear why “paired theme...” is not provided with a number, but the following subheading is	Thank you, this is an important point. We have reorganised the Results section to take into consideration your comment. Hope the new layout is easier to follow.
My overall suggestion for the results section is to try reduce this from (almost) 8 pages of text to 5-6 – it is simply too much info / too long otherwise, even if this is a qualitative paper. I guess you can reduce text particularly in the description of the individual themes.	Thank you for your comment. We have cut down the words in the Results section.
Readers will tend to look at figure one without reading the text beforehand and I am not sure if this figure is self-explanatory, particularly re where it starts, where it ends and at which point the actual injury happens / can be imagined. The 3 outer boxes (catastrophic expenditure etc) are unclear, how do they relate to the cycle?	Thank you for your comment. We have reflected on this comment and feel that the figure is not adding anything to our results. We have therefore decided to remove it from the manuscript.
Discussion The very first paragraph should entail a summary with the main barriers and facilitators and a (very) short explanation of each. Now it seems unclear where this summary section ends and if you also	We have added the summary of findings at the beginning of the Discussion section.

describe facilitators and not just barriers?	
Will be very helpful to insert thematic sub-headings for the first 3 pages of the discussion for greater clarity/structure	We have rewritten the Discussion to add clarity and strengthen our arguments.
Conclusion: Would be good to, again, say what hinders and facilitates good injury care most so far and what kind of “effective strategies” you would really suggest – maybe also by considering the local situation? This can still be concise as you don’t need to make the manuscript longer	Thank you for your comment. We have changed the conclusions to take into account your comment.

VERSION 2 – REVIEW

REVIEWER	Julia Hews-Girard University of Calgary, Faculty of Nursing
REVIEW RETURNED	15-May-2024

GENERAL COMMENTS	Thank you for all the work addressing the previous comments.
--

REVIEWER	Jonas Lander Hannover Medical School, Institute for Epidemiology, Social Medicine and Health Systems Research
REVIEW RETURNED	14-May-2024

GENERAL COMMENTS	Dear editor and authors, I have looked at the track changes in the documents and read the answers to my previous questions. I feel that the authors have addressed these points sufficiently and thus have no further questions at this point. As I am not a native speaker, I am not entirely in a position to judge the style of writing, but some of the co-authors seem to be native speakers indeed. Kind regards, reviewer 2
---